# Polyphenol and Anthocyanin Composition and Activity of Highland Barley with Different Colors

**DOI:** 10.3390/molecules27113411

**Published:** 2022-05-25

**Authors:** Hui-Min Jin, Bin Dang, Wen-Gang Zhang, Wan-Cai Zheng, Xi-Juan Yang

**Affiliations:** 1Academy of Agricultural and Forestry Sciences, Qinghai University, Xining 810016, China; 19083203130@qhu.edu.cn (H.-M.J.); dangbin811@tom.com (B.D.); zhangwgang0402@sina.com (W.-G.Z.); 13565849218@163.com (W.-C.Z.); 2Tibetan Plateau Key Laboratory of Agric-Product Processing, Qinghai Academy of Agricultural and Forestry Sciences, Xining 810016, China; 3College of Food Science and Engineering, Northwest A & F University, Xianyang 712100, China; 4Laboratory for Research and Utilization of Qinghai Tibet Plateau Germplasm Resources, Qinghai Academy of Agricultural and Forestry Sciences, Xining 810016, China

**Keywords:** highland barley, polyphenol composition, antioxidant activity, alpha-glucosidase activity inhibition

## Abstract

In this research, the composition of free phenols, bound phenols, and anthocyanins and their in vitro antioxidant activity and in vitro α-glucosidase inhibiting activity were observed in different barley colors. The outcomes revealed that the contents of total phenols (570.78 mg/100 gDW), total flavonoids (47.08 mg/100 gDW), and anthocyanins (48.07 mg/100 g) were the highest in purple barley. Furthermore, the structure, composition, and concentration of phenolics differed depending on the colors of barley. The types and contents of bound total phenolic acids and flavonoids were greater than those of free total phenolic acids and flavonoids. The main phenolic acids in blue barley were cinnamic acid polyphenols, whereas in black, yellow, and purple barley, benzoic acid polyphenols were the main phenolic acids, and the main types of flavonoids in black and blue barley were chalcones and flavanones, respectively, whereas flavonol was the main type of flavonoid in yellow and purple barley. Moreover, cornflower pigment-3-glucoside was the major anthocyanin in blue, yellow, and purple barley, whereas the main anthocyanin in black barley was delphinidin-3-glucoside. The dark color of barley indicated richness in the anthocyanins. In addition, the free polyphenol fractions had stronger DPPH and ABTS radical scavenging capacity as compared to the bound ones. In vitro α-glucosidase-inhibiting activity was greater in bound polyphenols than in free polyphenols, with differences between different varieties of barley. Purple barley phenolic fractions had the greatest ABTS radical scavenging and iron ion reduction capacities, as well as the highest α-glucosidase-inhibiting activity. The strongest DPPH radical scavenging capacity was found in yellow barley, while the strongest in vitro α-glucosidase-inhibiting activity was found in anthocyanins isolated from black barley. Furthermore, in different colors of barley, there was a strong association between the concentration of specific phenolic compounds and antioxidant and α-glucosidase-inhibiting activities. The outcomes of this study revealed that all colored barley seeds tested were high in phenolic compounds, and had a good antioxidant impact and α-glucosidase-inhibiting activity. As a result, colored barley can serve as an antioxidant and hypoglycemic food. Polyphenols extracted from purple barley and anthocyanins extracted from black barley stand out among them.

## 1. Introduction

Highland barley (barley), also known as hulless barley, is mainly distributed on the Qinghai-Tibet Plateau at an altitude of approximately 1400–4700 m, having a 3500-year cultivation history. It is the most widely grown crop in Tibetan areas of China and the main food for Tibetan farmers and herders; thus, it is one of the most important crops in the Tibetan Plateau region [1,2]. Barley is gaining popularity due to its high protein, fiber, and vitamin content and low fat and sugar properties along with its high content of functional components [3]. Long-term consumption of barley prevents diseases like diabetes, hyperlipidemia, and atherosclerosis as shown by epidemiological studies [1]. Consequently, barley can be used as a functional food ingredient and it can also be a source of natural antioxidants [4]. The main bioactive components in barley are β-glucan and phenolics [5]. Barley produces enriched phenolic compounds under alpine, anoxic, and high-light climatic conditions on the Qinghai-Tibet Plateau. Hence, barley is a rich source of phenolic compounds and an interesting research topic [6,7].

Phenolic compounds are an essential secondary metabolite of plants formed when a hydroxyl group replaces the hydrogen atom on the benzene ring of an aromatic hydrocarbon. They are very potent antioxidants that scavenge reactive-free radicals and chelate iron, and also can inhibit lipid peroxidation [8,9,10]. Phenolic compounds can be divided into monophenols and polyphenols based on the number of hydroxyl and phenolic groups contained in the molecule; Polyphenols can be further divided into flavonoids, stilbene, lignans, and phenolic acids [7]. Barley contains 0.1–0.4% phenolic compounds (benzoic acids, cinnamic acids, proanthocyanidins, benzoquinones, flavonols, chalcones, flavones, flavanols, and aminophenols), which is a much higher proportion than that of other cereals. These compounds are mainly present in free or bound phenolic fraction, with 80% present in the bran and endosperm [11]. Due to their strong capacity to scavenge free radicals [12,13], they can lower blood lipids and sugar, prevent cardiovascular diseases, improve human immunity, and exert antiviral and anticancer effects [3,14].

China is rich in barley varieties, which are available in different colors and shapes. Colored barley is a valuable germplasm resource, mainly including black barley, purple barley, and blue barley. According to a previous study, colored barley is rich in natural pigments and has a higher content of phenolic compounds, such as polyphenols, flavonoids, and anthocyanins than ordinary barley [5]; therefore, the development of colored barley as a functional food has been receiving increased attention. Furthermore, the type, content, and activity of phenolic compounds contained in barley are influenced by the seed color [12,13,14]. However, most existing studies have focused on hull barley, while hulless barley has been rarely studied. The existing reports have highlighted the composition, content, composition, and antioxidant activity of phenolic compounds extracted from colored barley, indicating it as a potential source of antioxidants and phenolic compounds [4,5,6]. Though, these studies have ignored the differences in the composition and content of anthocyanin-like components in different colored barley. Moreover, the differences in composition and content of free and bound phenols, and their antioxidant and α-glucosidase-inhibiting activities based on different barley colors were not reported. Consequently, the scientific basis for the evaluation of healthcare functions of different colored barley is insufficient and unable to guide researchers to study colored barley.

Therefore, in this study, we selected representative barley varieties of different colors cultivated on the Tibetan plateau to compare the composition and content, as well as in vitro antioxidant activity and α-glucosidase inhibiting capacity, of free and bound phenolic compounds and anthocyanin in barley of different colors. The study results provide a theoretical basis for enriching the functional chemical species and health effects of colored barley, improving people’s understanding of the crop and promoting its development and utilization.

## 2. Results and Discussion

### 2.1. Content of Phenolic Compounds in Barley

As shown in Table 1, the phenolic acid content in the different colored barley was approximately 10 times higher than the flavonoid and anthocyanin contents, indicating that phenolic acids are the main phenolic compounds in barley. The ranking (from highest to lowest) of the total phenolic content in the four barley species was as follows: purple (570.78 ± 18.83 mg/100 g DW) > blue (542.28 ± 26.51 mg/100 g DW) > yellow (515.72 ± 18.57 mg/100 g DW) > black (509.44 ± 15.76 mg/100 g DW). The purple barley had the highest total phenolic compound content. Interestingly, all four barley varieties exhibited higher total phenolic compound content than the values reported by Lin et al. [5]. In addition, the bound phenol content of all four species was significantly higher than the corresponding free phenol content, indicating that bound phenolic compounds were the main fraction of polyphenols present in colored barley, which is consistent with the results of Yang et al. [6]. The four barley varieties were ranked (from highest to lowest) by the bound phenol content as black > blue > purple > yellow, whereas the free phenol content was ranked as purple > blue > yellow > black. In contrast, the ranking based on free phenol content was purple > blue > yellow > black. Black barley exhibited the highest bound phenol content (344.35 ± 13.17 mg/100 g DW). In contrast, it had the lowest free phenol content (165.09 ± 5.17 mg/100 g DW), less than half compared to the bound fraction. Purple barley had the highest free phenolic compound content among all varieties tested with 264.26 ± 19.14 mg/100 g DW. The difference in free phenolic compounds content between the yellow and blue varieties was insignificant. Overall, these results are consistent with those obtained by Lin [5], suggesting that both blue and black barley had higher levels of free and bound phenolic compounds than yellow barley.

The total flavonoid content of purple, black, blue, and yellow colored varieties were 47.08 ± 6.17, 45.53 ± 4.56, 38.47 ± 2.97, and 38.04 ± 4.34 mg/100 g DW, respectively. Except for the yellow barley, the differences in total flavonoid content among the other barley varieties were not statistically significant. These findings agree with previous studies where the flavonoid content of dark barley was higher than that of yellow barley [5]. Additionally, the free flavonoid content of all four colors of barley was significantly higher than the corresponding bound flavonoid content, indicating that the flavonoids in barley existed mainly in the free phenolic fraction, which is consistent with the results of Zhu et al. [15]. According to the free flavonoid content, the four varieties were ranked (from highest to lowest) as black > purple > blue > yellow, whereas the bound flavonoid content was ranked as purple > black > yellow > blue. The free flavonoid content in black barley was 29.93 ± 2.64 mg/100 g DW, which was significantly higher than those of the other barley species. The bound flavonoid content in purple barley was the highest (19.33 ± 5.64 mg/100 g DW), which was not significantly different from that of the black variety but significantly higher than those of the blue and yellow varieties.

The anthocyanin contents of the purple, black, yellow, and blue varieties were 48.07 ± 8.73, 41.91 ± 0.13, 38.09 ± 0.06, and 33.53 ± 5.07 mg/100 g, respectively. Except for the blue variety, the differences in anthocyanin contents of the other three varieties were not significant and were similar to the results of Lin et al. [5]. Himi et al. found that the grain color of colored barley is significantly related to phycocyanin synthesis [16,17,18]. In addition, the content and fraction of phenolic compounds in different colored barley were found to be strongly influenced by the seed color. Another study suggested that colored barley is richer in phenolic compounds and that barley contains unique phenolic compounds with potential health benefits [4].

### 2.2. Composition and Content of Phenolic Compounds in Barley

#### 2.2.1. Composition and Content of Phenolic Acids and Flavonoids in Barley of Different Colors

In this study, the phenolic acids such as phloroglucinol, gallic acid, protocatechuic acid, chlorogenic acid, 2, 4-Dihydroxybenzoic acid, vanillic acid, syringic acid, p-coumaric acid, ferulic acid, salicylic acid, benzoic acid, o-coumaric acid, and 3, 4-Dimethoxybenzoic acid, and flavonoids such as catechin, naringin, hesperidin, myricetin, quercetin, naringenin, kaempferol, and rutin were detected in the free and bound phenolic extracts of blue highland barley, using a high-performance liquid chromatography (HPLC) system (Figure 1). The composition and content of phenolic acids and flavonoids in barley of different colors was in Table 2. There were some differences in the composition and content of phenolic acids and flavonoids in the different varieties of barley. We detected 14, 12, 14, and 12 types of phenolic acids and 14, 12, 10, and 16 types of flavonoids in the black, yellow, blue, and purple varieties, respectively. These results indicate that colored barley is rich in phenolic compounds, which is consistent with previous findings [4,19]. In this study, bound phenolic acids and flavonoids were more abundant in barley than free ones, and the contents of total bound phenolic acids and flavonoids were higher than that of free phenolic compounds. The total phenolic acid content was highest in the blue variety (611.605 ± 2.656 μg/g), which was 2.94, 1.92, and 6.33 times higher than those of black, yellow, and purple varieties, respectively. The total flavonoid content was also highest in the blue variety (106.442 ± 1.06 μg/g), which was 1.5, 2.48, and 1.89 times higher than those in the black, yellow, and purple varieties, respectively. The polyphenol and flavonoid contents varied considerably in different barley varieties, which is consistent with the findings of Lin et al. [5]. However, the concentrations reported are inconsistent with those obtained with the chemical method we employed in this study. We hypothesized that the discrepancies are due to calculation methodology differences.

The phenolic acids in the blue variety were mainly dominated by cinnamic acids, accounting for 97.52% of the total phenolic acid content. In addition, p-coumaric acid had the highest content (583.539 ± 2.248 μg/g), which was only detected in the bound fraction. The phenolic acids in the black, yellow, and purple varieties were all dominated by benzoic acids, accounting for 87.80%, 93.26%, and 88.33% of the total, respectively. Syringic acid (25.638–116.901 μg/g) and benzoic acid (29.596–140.898 μg/g) had the highest contents, both of which were in the free phenolic fraction. These results are inconsistent with those reported by El-Saye et al. [4], according to which ferulic acid is the dominant phenolic acid in barley, wheat, and rye, which might be related to the genotype, growing environment, and extraction method. Salicylic acid was only detected in the free phenolic compounds of the four varieties, while syringic acid, salicylic acid, benzoic acid, veratric acid, p-coumaric acid, ferulic acid, and o-coumaric acid were all present in the bound phenolic acid extracts of the four varieties. Additionally, vanillic acid was only detected in the black and blue varieties, whereas gallic acid was only detected in the yellow and purple varieties. These findings suggested that the total phenolic compounds content (sum of free and bound phenolic compounds) was higher in barley than in maize, wheat, oats, rice [20], and pearl barley [21]. In addition, the total phenolic compounds content in barley was higher than those in other barley species [22]. These findings suggest that, in the Tibetan plateau, barley has a higher phenol acid content and is more suitable for exploitation.

In this study, the flavonoids in barley were mainly present in the bound fraction, which is consistent with the results of Adom [23], who found that the bound flavonoid content in wheat was higher than the corresponding free flavonoid content. Wang [22] reported that the total flavonoid content of the bound phenolic fraction was higher than the corresponding free fraction. The highest total flavonoid content was detected in the blue variety (106.442 ± 1.06 μg/g), followed by the black (70.796 ± 0.898 μg/g), purple (56.206 ± 0.336 μg/g), and yellow (42.944 ± 0.906 μg/g) varieties. Among them, the dominant type of flavonoids in the black variety was chalcone, which accounted for 48.2% of the total flavonoids; phloretin was the most abundant compound, and its free phloretin content (33.078 ± 0.164 μg/g) was much higher than that of the bound one (1.047 ± 0.073 μg/g); hesperidin was the most abundant compound among the bound flavonoids in the black variety (5.564 ± 0.26 μ g/g), but it was not detected in the free fraction; the main type of flavonoids in the yellow variety was flavonols, accounting for 53.88% of total flavonoids, and the bound flavonol content was 3.76 times higher than that of the free one. Catechin had the highest content (5.693 ± 0.198 μg/g) of free flavonoids, which was not detected in the bound flavonoids. Naringin (8.066 ± 0.115 μg/g) and quercetin (8.055 ± 0.129 μg/g) were the most abundant of the bound flavonoids; flavanones and flavonols were the main types of flavonoids in the blue and purple varieties, accounting for 49.58% and 43.87% of the total flavonoids in the blue variety, respectively, and were mainly present in the bound phenolic fraction. Naringenin was the most abundant compound, with 50.145 ± 1.156 μg/g content in the bound phenolic fraction and 0.625 ± 0.007 μg/g content in the free phenolic fraction, followed by quercetin (24.798 ± 1.168 μg/g), which was only present in the bound fraction. The proportions of flavanones and flavonols in the purple variety were 38.77% and 40.39%, respectively, which were predominately bound flavonoids. The quercetin content was higher in the free phenolic fraction (8.356 ± 0.977 μg/g) than in the bound phenolic fraction (3.729 ± 0.109 μg/g); hesperidin and naringenin were the most abundant flavonoids in the bound phenolic fraction (9.688 ± 0.059 and 8.547 ± 0.151 μg/g, respectively), which were also higher than those in the free phenolic fraction (2.856 ± 0.074 and 0.698 ± 0.018 μg/g). Naringin, naringenin, hesperidin, myricetin, quercetin, kaempferol, and rutin were all present in the bound flavonoids of the four varieties; naringenin and rutin were detected in the free fraction flavonoids of all four varieties, consistent with the finding that the flavonoids contained in barley include quercetin, rutin, and kaempferol [4]. In addition, phloretin was only detected in the black and purple varieties; catechins were present in black, yellow, and purple varieties and were not detected in the blue variety. These results indicate that flavonoid fraction and their compound composition varied significantly among the differently colored barley varieties. Free phloretin was the main flavonoid compound in the black-colored variety, while bound naringenin and quercetin were abundant in the blue-colored variety. The dominant flavonoids in the yellow-colored variety were free catechin, bound naringin, and quercetin. Finally, the purple-colored variety had a higher content of free quercetin, bound hesperidin, and naringenin. These findings provide reference points for phenolic acid and flavonoid screening and quality evaluation in different colored barley varieties.

#### 2.2.2. Composition and Content of Anthocyanin Compounds in Barley of Different Colors

Barley varieties with different colors have been reported to be rich in anthocyanins, which have a positive effect in reducing the risk of cardiovascular disease [5,24]. The total ion chromatogram of anthocyanins and LC-MS chromatogram of anthocyanin standard were in Figure 2 and Figure 3, and the anthocyanin composition and content of the four barley varieties are shown in Table 3. The purple variety had the highest total anthocyanin content (105.213 ± 2.514 μg/g), followed by the blue (27.422 ± 0.635 μg/g), black (8.492 ± 0.83 μg/g), and yellow varieties (4.413 ± 0.442 μg/g). Cyanidin, cyanidin-3-glucoside, delphinidin, pelargonidin-3-glucoside, petunidin, and petunidin-3-glucoside were the main types of anthocyanins in all barley varieties. Anthocyanins in cereals were mainly concentrated in the bran of the grain. Various anthocyanin compounds were identified in barley grain, including Dp-3-glucoside, Cy-3-glucoside, Cy-3-(6″-succinyl)glucoside, and Pn-3-(6-succinyl)glucoside, with a wide variation in composition [25]. Delphinidin-3-glucoside was the most abundant compounds among the anthocyanins in the black variety (3.013 ± 0.179 μg/g), while peonidin was not detected in the black variety. The most abundant compounds among the anthocyanins in the yellow variety were cyanidin-3-glucoside and delphinidin (1.267 ± 0.013 and 1.267 ± 0.051 μg/g, respectively), whereas malvidin-3-glucoside was only detected in the yellow variety. Cyanidin-3-glucoside (10.659 ± 0.088 μg/g) and petunidin-3-glucoside (6.004 ± 0.196 μg/g) had the highest contents among anthocyanins in the blue variety, accounting for 38.87% and 21.89% of the total anthocyanins, respectively. Cyanidin-3-glucoside was the most dominant compounds in the anthocyanins of the purple variety (96.874 ± 0.783 μg/g), accounting for 92.07% of the total anthocyanins. Delphinidin-3-glucoside was not detected in purple barley. Thus, our results demonstrated that the pigment compounds of the different barley varieties were mainly anthocyanins (99.761–1.558 μg/g). Their specific compounds differed depending on barley color: cyanidin-3-glucoside was the dominant anthocyanin in the yellow, blue, and purple colored varieties. Delphinidin-3-glucoside was uniquely dominant in the black-colored variety. Naturally colored foods are richer in phenolic compounds; the darker color of the beans indicates a higher content of polyphenolic compounds [2,5]. This agrees with our results, with the yellow-colored variety containing the lowest anthocyanin content. However, the purple and blue colored varieties were richer in anthocyanins and not the darker colored black variety.

### 2.3. In Vitro Antioxidant Activity of Phenolic Fractions from Different Varieties of Barley

#### 2.3.1. In Vitro Antioxidant Activity of Phenolic Acids and Flavonoids

As shown in Figure 4, in the three antioxidant assays, the phenolic fractions of different barley varieties had DPPH · and ABTS · scavenging ability and strong iron ion reduction capacity. According to the literature, phenolic compounds, because of their unique molecular structure, exhibit strong antioxidant and chelating properties and can scavenge free radicals in the body and exert an antioxidant effect [14,26]. The barley extract free fraction exhibited higher DPPH · and ABTS · scavenging ability than the bound fraction. The free fraction extracts had a stronger iron ion reduction capacity than the bound extracts in all varieties, except for the black-colored variety. This result suggests that the in vitro antioxidant capacity of the free phenolic compounds was stronger, which is consistent with a previously reported result that free phenolic compounds in buckwheat are the main contributor to the total free radical scavenging capacity [20]. This is due to the different compositions and contents of monomeric polyphenols in free and bound phenolic compounds in barley [4]. In addition, there were significant differences in the scavenging capacity of DPPH · for the different barley colors. The yellow variety showed the strongest DPPH · scavenging ability, followed by the blue and purple varieties, while the purple variety showed the strongest ABTS · scavenging ability, followed by the blue and yellow varieties. All these four varieties showed strong iron ion reduction ability, with the purple variety being the strongest, whereas the difference in the iron ion reduction ability among black, yellow, and blue varieties was not significant. The phenolic compounds in barley extracts exhibited higher antioxidant activity compared to wheat [27] and oats [28]. Lin [5] found that the black variety had the highest average antioxidant capacity, followed by the blue and white varieties. Ge [7] reported that the white variety had the strongest DPPH radical scavenging activity (13.24–60.09 mmol Trolox/100 g DW), followed by the black (22.44–51.76 mmol Trolox/100 g DW) and yellow (11.24–51.97 mmol/100 g DW) varieties. Additionally, the white and black varieties exhibited a stronger ability to reduce iron ions. These data further indicate on differences in the compositions and contents of phenolic compounds in different varieties of barley. Moreover, different monomeric phenolic compounds are selective for scavenging different free radicals, which yields differences in their activity [4,6,29].

#### 2.3.2. In Vitro Antioxidant Activity of Anthocyanin Fractions

As shown in Figure 5, anthocyanin fractions from all four varieties had some in vitro antioxidant capacity. Among them, the iron ion reduction ability was the strongest, followed by the ability to scavenge ABTS ·, and the ability to scavenge DPPH · was relatively weak. By consuming functional foods containing natural antioxidant compounds such as anthocyanins, excess free radicals can be scavenged and the dynamic balance of the body’s internal environment can be maintained [30]. Anthocyanin fractions from the four different varieties had different antioxidant activity, depends of applied antioxidant assays: the purple variety had the strongest ABTS · scavenging ability, followed by the black and yellow varieties; the purple variety had the strongest iron ion reduction capacity, followed by the black and yellow varieties. This result indicates that the purple variety had strong iron ion reduction capacity and ABTS · scavenging ability, which is consistent with the results of Zhang [31] who reported that anthocyanins extracted from the purple variety had high iron ion reduction capacity and ABTS · scavenging ability. The ranking (from strong to weak) of the DPPH · scavenging ability of different varieties was yellow > black > blue > purple, with significant differences. However, this finding is inconsistent with that reported by Kim [32], according to which the DPPH · scavenging ability of anthocyanin extracts from the purple variety was higher (67.4%) than that of the black variety (63.5%). This is attributable to varietal and environmental differences. In addition, this finding is inconsistent with that obtained by Hu et al. [33], who reported that the DPPH · scavenging ability of anthocyanins was higher in black maize variety than in white and yellow glutinous varieties, which is attributable to the different types and contents of anthocyanins contained in different crops.

### 2.4. Inhibition of Alpha-Glucosidase Activity In Vitro

Alpha-glucosidase is a key enzyme involved in the digestion and degradation of carbohydrates in the small intestine, which can degrade the carbohydrates ingested by the body into glucose. Alpha-glucosidase inhibitors can reduce glucose production by inhibiting α-glucosidase activity, slowing down the rate of glucose entering the blood and, thus, reducing the postprandial blood glucose rise and stabilizing the body’s blood glucose level [34,35]. The results of in vitro inhibition of α-glucosidase by polyphenol fractions from barley of different colors are shown in Figure 6. All these varieties showed a dose-dependent relationship for the inhibition of α-glucosidase activity, which is consistent with the findings of Ervina [36]. The IC50 values of α-glucosidase in polyphenols decreased with increasing polyphenol content. Moreover, the IC50 values for the in vitro inhibition of α-glucosidase by the free fraction of barley, calculated by SPSS software, were 15.881, 23.904, 8.624, and 8.624 mg/g for the black, yellow, blue, and purple varieties, respectively. The IC50 value of α-glucosidase was 9.110 mg/g using acarbose as the control group, and the free polyphenols in the purple and blue varieties had a stronger inhibitory effect than acarbose. Additionally, the IC50 values for the in vitro inhibition of α-glucosidase by the bound fractions were 4.923, 3.340, 2.102, and 3.369 mg/g for the black, yellow, blue, and purple varieties, respectively, all of which were stronger than those of acarbose. In addition, the IC50 values of anthocyanin fractions were 7.397, 8.298, 9.36, and 11.263 mg/g for the black, yellow, blue, and purple varieties, respectively; the inhibition of α-glucosidase in vitro by anthocyanins in the black and yellow varieties was stronger than that of acarbose. Pradeep [37] confirmed that plant phenolic compounds can inhibit α-glucosidase activity and significantly reduce the rate of starch hydrolysis. Bellesia [38] found that phenolic compounds can inhibit the activity of α-glucosidase. The inhibitory effect of procyanidin B2 (IC50: 0.19 mg/mL) was stronger than those of epicatechin (IC50: 0.27 mg/mL), procyanidin B1 (IC50: 0.29 mg/mL), and catechin (IC50: 0.38 mg/mL). The results of our study suggested that polyphenols in the purple and blue varieties exhibited the strongest inhibition of α-glucosidase in vitro, whereas anthocyanins in the black variety exhibited the strongest inhibition of α-glucosidase.

### 2.5. Correlation Analysis

In this research, a correlation analysis was done to understand the link between phenolic compounds extracted from barley and their in vitro activity in more detail. Table 4 highlights a strong positive link between free total phenolic acid content and capacity of iron ion reduction (*p* < 0.05), suggesting that free phenolic fractions from various colored barley have more phenolic compounds with iron ion reduction capacity. Lin’s finding that total phenolic acids in different colored barley (black, blue, and white) had a strong positive connection with iron ion reduction capacity is consistent with this conclusion [5]. Total cinnamic acids, o-coumaric acid, vanillic acid, and phloretin all demonstrated a substantial positive connection with FRAP reduction capacity (*p* < 0.05), implying that they were the primary contributors to iron ion reduction capacity. Iron ion reduction capacity had a significant negative connection with protocatechuic acid and chlorogenic acid (*p* < 0.05), showing that different forms of free phenols are selective for different methods of antioxidant activity evaluation. This conclusion contradicts Ge’s findings [7], which showed a substantial positive association between phloretin and DPPH ·, as well as coumaric acid and ABTS ·. Gallic acid in bound phenolic compounds had a strong negative connection with the ABTS · radical scavenging capacity (*p* < 0.05), similar to Zhao [39] and Yang [6]. Benzoic acid demonstrated a significant positive association (*p* < 0.05) with DPPH radical scavenging capacity, showing that benzoic acid was the primary contributor to DPPH-radical scavenging by bound phenols in various colored barley. This, however, contradicts Yang’s discovery that quercetin was the predominant contributor to DPPH-radical scavenging by bound phenolic fractions from blue barley [6]. The half-maximal inhibitory concentration (IC50) of pelargonidin-3-glucoside in anthocyanins was strongly linked with the α-glucosidase-inhibiting activity, highlighting that a greater content of pelargonidin-3-glucoside caused weaker α-glucosidase-inhibiting activity. Petunidin had a strong negative link with the α-glucosidase-inhibiting activity (−0.615). Consequently, elevated petunidin content led to stronger α-glucosidase-inhibiting activity, suggesting that petunidin was the key contributor to the α-glucosidase inhibition by anthocyanins from various colored barley.

Although numerous compounds in the current investigation had high correlation coefficients with antioxidant and α-glucosidase-inhibiting activities, few of them achieved the significant correlation threshold, which could be due to the small number of samples included in the study (4). As a result, the sample size should be increased for subsequent research. Furthermore, antioxidant and α-glucosidase-inhibiting activities were substantially connected with compounds such as o-coumaric acid, vanillic acid, pelargonidin-3-glucoside, and petunidin, however, their levels were not high in the barley evaluated in this study. As a result, content is not the only thing that influences their activity; the impact of their structure on the activity should be investigated as well [7]. The types of chemicals that were substantially connected with the activity of phenolic fractions from different colored barley in this study were not the same as those previously reported [5,6,7], which could be due to the diverse barley varieties used and their growing regions. As a result, genotype and cultivation environment may have a substantial impact on the types and quantities of phenolic compounds in different colored barley, as well as their antioxidant and α-glucosidase-inhibiting properties.

## 3. Materials and Methods

### 3.1. Material and Reagents

Kunlun 20 (black variety), Kunlun 15 (yellow variety), Ganqing 4 (blue variety), and Yunqing 2 (purple variety) highland barley were planted in the same experimental plot of the Qinghai Academy of Agricultural and Forestry Sciences (Xining), and were used after harvest in 2021(Figure 7). 1,1-diphenyl-2-picrylhydrazyl (DPPH), Tri-2-pyridyl-s-triazine (TPTZ) Trolox (water-soluble vitamin E), ABTS, α-glucosidase, and p-nitrophenol glucopyranoside (PNPG) were supplied by Sigma-Aldrich in St. Louis, MO, USA. Gallic acid, rhizobioside, protocatechuic acid, chlorogenic acid, 2,4-dihydroxybenzoic acid, vanillic acid, eugenol, p-coumaric acid, ferulic acid, salicylic acid, benzoic acid, quinoa, o-coumaric acid, rutin, naringin, hesperidin, myricetin, quercetin, naringenin, and kaempferol standards (purity ≥ 98.0%) were supplied by Shanghai Yuanye Biotech. Folin-phenol (guarantee reagent) was supplied by Beijing Solarbio. Sodium carbonate, sodium hydroxide, potassium chloride, sodium acetate, sodium nitrite, aluminum nitrate, ferric chloride, and potassium persulphate were supplied by Tianjin Hengxing Chemical Reagent. Disodium hydrogen phosphate, sodium dihydrogen phosphate, and 3,5-dinitrosalicylic acid were supplied by Sinopharm Chemical Reagent. Acetone, methanol, n-hexane, ethyl acetate, and concentrated HCl were provided by Tianjin Fuyu Fine Chemical. Moreover, the water used in this study was deionized water…

### 3.2. Research Procedures

#### 3.2.1. Extraction of Free Phenolic Compounds from Barley

Using Zhao’s [39] method with slight modification, 1.0 g of barley powder was added to 80% acetone by volume in a ratio of 1:25. Subsequently, the samples were extracted by ultrasonication at room temperature for 30 min and centrifuged at 4000 r/min for 10 min. After collecting the supernatant, the residue was subjected to the above treatment two times using the same method. The supernatant obtained in the three procedures was mixed, rotary-dried at 45 °C under reduced pressure, and dissolved in 10 mL methanol, to obtain the extracts of free phenolic compounds. Finally, they were stored at −20 °C away from light after subpackaging. The entire procedure was repeated three times.

#### 3.2.2. Extraction of Bound Phenolic Compounds from Barley

According to Yang’s [6] method, 20 mL of n-hexane was added to the residue after the extraction of free phenolic compounds. After shaking, the supernatant was discarded by centrifugation at 3000 rpm for 5 min. Then, the precipitate was added to 17 mL of 11.00% *v/v* sulfuric acid in a water bath at 75 °C for 1 h. Afterward, the precipitate was extracted five times with 20 mL of ethyl acetate and centrifuged at 3000 rpm for 5 min. The ethyl acetate extract phases were combined and, subsequently, evaporated to dryness by rotation at 45 °C. The residue was methanol-fixed to 10 mL and filtered through a 0.45-μm-pore organic membrane to obtain the bound phenol extract. Finally, the extracts were stored at −20 °C away from light after subpackaging.

#### 3.2.3. Extraction of Anthocyanins from Barley

One gram of barley powder was added to 10 mL of 0.1% HCl methanol solution, soaked for 15 min, ultrasonicated at 100 Hz for 30 min at room temperature, and centrifuged at 4000 rpm for 20 min. After the supernatant was collected, the residue was extracted two times using the same method. The supernatant obtained in the three procedures was mixed and dried by rotary evaporation at 45 °C under reduced pressure. To obtain anthocyanin extract, the precipitate was dissolved in 10 mL water and filtered through a 0.45-μm-pore organic membrane. Finally, the extracts were stored at −20 °C away from light.

#### 3.2.4. Determination of Active Ingredients

Determination of total phenolic compounds

For determining total phenolic compounds, we adopted the Folin-Ciocalteu reagents assay method and referred to Adom’s [23] method with slight modification. After aspirating 125 μL of the sample extract into a test tube, 500 μL of distilled water and 125 μL of folin-phenol were added and shaken well. After 6 min of reaction, 1.25 mL of 7% Na_2_CO_3_ solution was added to the test tube, followed by the addition of 1 mL of distilled water. As a blank solution for zeroing the spectrophotometer, the sample extraction solution was replaced by methanol. The absorbance value was measured after 1.5 h storage at room temperature away from light, at a wavelength of 765 nm. All samples and measurements were performed in triplicates. Gallic acid was used as the standard, the total phenol content was calculated according to the standard curve (Y = 0.0042X + 0.0124 (0–300 μg/mL, R_2_ = 0.9996)), and the total phenolic contents were expressed as mg gallic acid equivalents (GAE)/100 g DW. 

2.Determination of total flavonoid content.

First, 100 μL of the sample extraction solution was added to the test tube. Subsequently, 200 μL of the 5% NaNO_2_ solution was added and shaken well. After 6 min of reaction, 200 μL of 10% Al(NO_3_)_3_ solution was added to the test tube and shaken well. After another 6 min reaction, 2 mL of 4% NaOH solution was added. The absorbance value was measured after 1.5 h storage at room temperature away from light, at a wavelength of 510 nm. All samples and measurements were performed in triplicates. The total flavonoid content was calculated according to the standard curve (Y = 0.0055X − 0.0047 [0–80 μg/mL, R_2_ = 0.9947]), and the total flavonoid contents were expressed as mg rutin equivalents (GAE)/100 g DW [6]. 

3.Determination of anthocyanin content.

Anthocyanin content was determined using pH-differential spectrophotometry [40]. The extraction solution was diluted with potassium chloride–hydrochloric acid buffer (0.2 mol/L) at pH 1.0 and sodium acetate–hydrochloric acid buffer (0.2 mol/L) at pH 4.5. After equilibration for 1 h, the absorbance was measured at 510 nm and 700 nm wavelengths three times in parallel. The anthocyanin content was calculated according to the following equation:Anthocyanin content (mg/100 g) = A × m/((εL) × Mw × DF × 100 × V)(1)
where A indicates absorbance; ε indicates the extinction coefficient of cyanidin-3-glucoside (26,900); L is the optical path (1 cm); Mw is the relative molecular mass of cyanidin-3-glucoside (449.2); DF is the dilution fold; V is the volume of the stock solution of the anthocyanin sample to be measured (mL); and m is the mass of barley after drying (g).
A = (A515 nm (pH 1.0) − A700 nm (pH 1.0)) − (A515 nm (pH 4.5) − A700 nm (pH 4.5))(2)

Distilled water was used as a control and A700 nm was used to eliminate the effect of turbidity in the sample solution.

#### 3.2.5. Polyphenol Composition in Barley

Composition and content determination of phenolic compounds in barley

High-performance liquid chromatography (HPLC) (WATERS 600E, DELTA 600 pump, Milford, CT, USA) was used to assess the composition of phenolic compounds extracted from the barley samples. Afterward, a Phenomenex C18 column (250 mm × 4.6 mm) with UV/VIS detector was employed to observe the samples at 280 nm with the system settings set as follows: run time, 60 min; injection volume, 20 μL; flow rate, 0.8 mL/min; mobile phase: distilled water containing 0.1% glacial acetic acid (solvent A) and acetonitrile containing 0.1% glacial acetic acid (solvent B). The settings for the elution gradient were as follows: 0 min, 8% B; 2 min, 10% B; 27 min, 30% B; 50 min, 90% B; 52–56 min, 100% B; and 56–60 min, 8% B. Individual peaks were detected and expressed as μg/g DW based on the retention time calculated for each pure compound [41].

2.Composition and Content Determination of Anthocyanins

Liquid chromatography-mass spectrometry (LC-MS/MS) was used to examine the composition of anthocyanins isolated from four different barley samples (Q-Exactive, from Thermo Fisher, New York, NY, USA). The parameters set for a Hypersil GOLD aQ column (100 mm × 2.1 mm, 1.9 µm) were as follows: mobile phase, distilled water containing 0.9% acetic acid (solvent A), and acetonitrile containing 0.9% acetic acid (solvent B). Moreover, the gradient elution settings were as follows: 0~0.5 min, 98% A and 2% B; 0.5~8 min, 2~50% B; 8~10 min, 50~90% B; 10~12 min, 90% B; 12~13 min, 2~90% B; 13~15 min, 2% B; flow rate, 0.30 μL/min; injection volume, 3µL. Subsequently, MS was carried out in positive ion mode with 100–1500 *m*/*z* scan range (full-scan mode, 70,000 resolution), 3.5 kV spray voltage, 300 °C capillary temperature, 300 °C heater temperature, 35 units/min sheath gas flow rate, and 10 units/min axillary gas flow rate.

#### 3.2.6. Determination of Antioxidant Capacity

Determination of DPPH’s radical scavenging capacity [42]

First, 1 mL of the extraction solution was mixed with 4.5 mL of 0.1 mmol/L DPPH–methanol solution. Then, the mixed solution was shaken well and placed for 30 min in the dark. As a blank solution for zeroing the spectrophotometer, the sample extraction solution was replaced by methanol. The absorbance values were measured at 517 nm. All samples and measurements were performed in triplicates. The DPPH · scavenging capacity was calculated based on the standard curve (Y = 0.0042X + 0.9163 (0–140 μmol/L, R^2^ = 0.9928)) using Trolox as the standard, which was expressed as Trolox equivalents per 100 g of extract (dry basis) (μmol/100 g).

2.Ferric reducing antioxidant power assay.

First, 1 mL of the extraction solution was mixed with 4.5 mL of FRAP working solution. Then, the mixed solution was shaken well and kept for 30 min away from light for the reaction to occur. As a blank solution for zeroing the spectrophotometer, the sample extraction solution was replaced by methanol. The absorbance values were measured at 593 nm. All samples and measurements were performed in triplicates. The Ferric reducing antioxidant power was calculated based on the standard curve (Y = 0.0072X − 0.0012 (0–300 μmol/L, R^2^ = 0.9992)) using Trolox as the standard, and the results were expressed as µmol TE/100 g DW [43].

3.Determination of radical scavenging capacity of ABTS ·

First, 200 mL of the extraction solution was mixed with 4 mL of ABTS · working solution. Then, the mixed solution was shaken well and underwent reaction for 30 min in the dark. As a blank solution for zeroing the spectrophotometer, the sample extraction solution was replaced by methanol. The absorbance values were measured at 734 nm. All samples and measurements were performed in triplicates. The ABTS · scavenging capacity was calculated based on the standard curve (Y = −0.001X + 0.6242 (0–300 μmol/L, R^2^ = 0.9907)) using Trolox as the standard, and the results were expressed as µmol TE/100 g DW [44].

#### 3.2.7. Inhibition of Alpha-Glucosidase Activity

Based on Yang’s [45] method with slight modification, 40 μL of phenolic compounds extraction solution (concentration from 0.5 mg/g to 10 mg/g) and 30 μL of α-glucosidase solution (0.2 U/mL) were added to a 96-well plate and incubated at 37 °C for 10 min after mixing well. Subsequently, 30 μL of PNPG (5 mmol/L) was added to each well and incubated for 30 min at 37 °C after mixing well. Finally, 100 μL of Na_2_CO_3_ (1 mol/L) was added to terminate the reaction. The absorbance values were measured at 405 nm and calculated as follows:Alpha-glucosidase solution inhibition rate/% = [1 − (A1 − A2)/A0] × 100(3)
where A0 is the absorbance value of the blank control, A1 is the sample absorbance value, and A2 is the background absorbance value of the phosphate buffer replacing the PNPG solution. The inhibition rate was calculated according to Equation (3), where the polyphenol concentration is the horizontal coordinate and the inhibition rate is the vertical coordinate. Finally, the polyphenol concentration at 50% inhibition (i.e., the half-inhibition concentration (IC50)) was calculated.

#### 3.2.8. Data Analysis

Data were analyzed and visualized using Excel and SPSS 17.0 software (SPSS Statistics 17.0.1, by Norman H. Nie, C. Hadlai Hull and Dale H. Bent in Chicago, USA), and the results were presented as mean ± standard deviation. Significant differences were determined using the LSD multiple-comparison method, and a correlation analysis was performed using the Pearson two-sided test.

## 4. Conclusions

The different colors of barley in this study were found to be rich in phenolic compounds. The form, composition, and content of the phenolics varied significantly depending on their color and variety. In general, the types and contents of phenolic acids and flavonoids in the bound phenolic fraction were higher than those in the total free fraction. The phenolic acids in the blue variety were mainly cinnamic acids, whereas those in the black, yellow, and purple varieties were mainly benzoic acids. The main type of flavonoid in the black variety was chalcone and that in the yellow variety was flavonol. Flavanones and flavonols were the most abundant in the blue and purple varieties; in all varieties, pigment compounds were mainly anthocyanins. Of these, cyanidin-3-glucoside was the characteristic anthocyanin in the yellow, blue, and purple varieties, whereas delphinidin-3-glucoside was the characteristic anthocyanin in the black variety. Additionally, the dark variety was richer in anthocyanins.

The phenolic fractions of the different barley varieties in this study showed strong DPPH · scavenging ability, ABTS · scavenging ability, iron ion reduction capacity, and α-glucosidase inhibition capacity. Among them, the free fraction polyphenol extracts showed stronger antioxidant capacity than the bound ones. However, the inhibition of α-glucosidase in vitro by the bound polyphenols was stronger than that by free ones, with differences among different barley varieties. The greatest contributions to the iron ion reduction capacity were o-coumaric acid, vanillic acid, and phloretin in the free phenolic compounds of different colored barley, whereas the main contributor to the DPPH radical scavenging capacity was benzoic acid in the bound phenolic compounds. Furthermore, petunidin was the primary inhibitor of α-glucosidase activity. This study’s findings could lead to the development of barley types that can be employed as natural antioxidants and food sources to prevent blood glucose increases in people. Animal investigations are needed to confirm the antioxidant and hypoglycemic properties of these barley varieties in vivo. The findings of this study can further improve the understanding of the phenolic compounds and their activities in different colored barley, thereby guiding the healthy consumption of barley and providing a theoretical basis for its development and utilization.

## Figures and Tables

**Figure 1 molecules-27-03411-f001:**
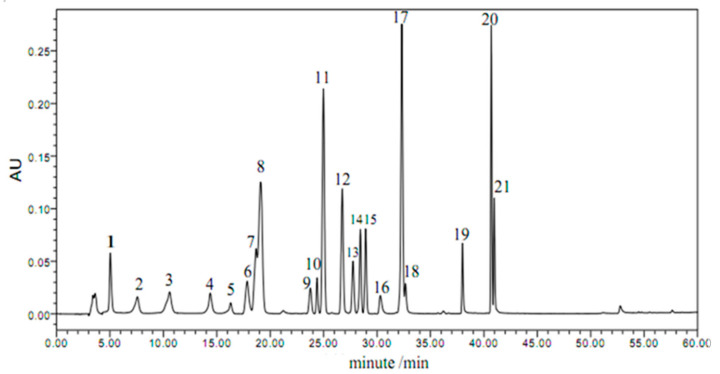
HPLC chromatogram of standards of phenolic compounds. (1) Phloroglucinol; (2) gallic acid; (3) protocatechuic acid; (4) chlorogenic acid; (5) catechin; (6) 2, 4-Dihydroxybenzoic acid; (7) vanillic; (8) syringic acid; (9) p-coumaric; (10) rutin; (11) ferulic acid; (12) salicylic acid; (13) naringin; (14) hesperidin; (15) benzoic acid; (16) o-coumaric acid; (17) myricetin; (18) quercetin; (19) 3, 4-Dimethoxybenzoic acid; (20) naringenin; (21) kaempferol.

**Figure 2 molecules-27-03411-f002:**
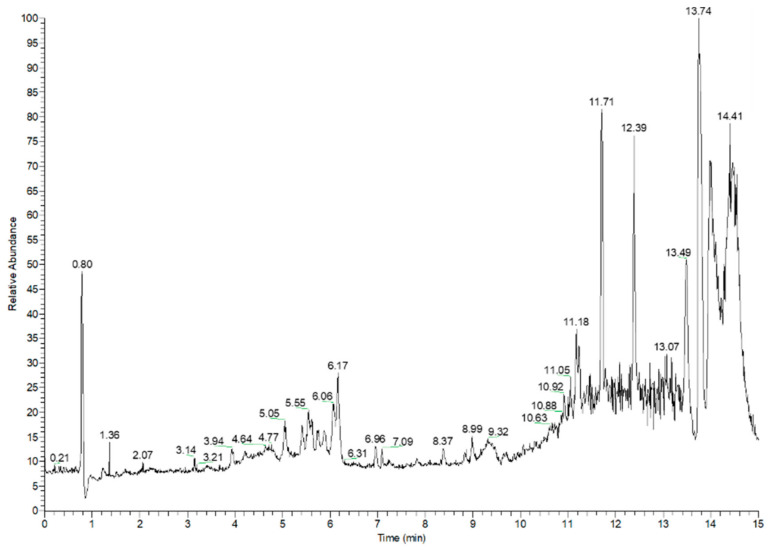
Total ion chromatogram of anthocyanins.

**Figure 3 molecules-27-03411-f003:**
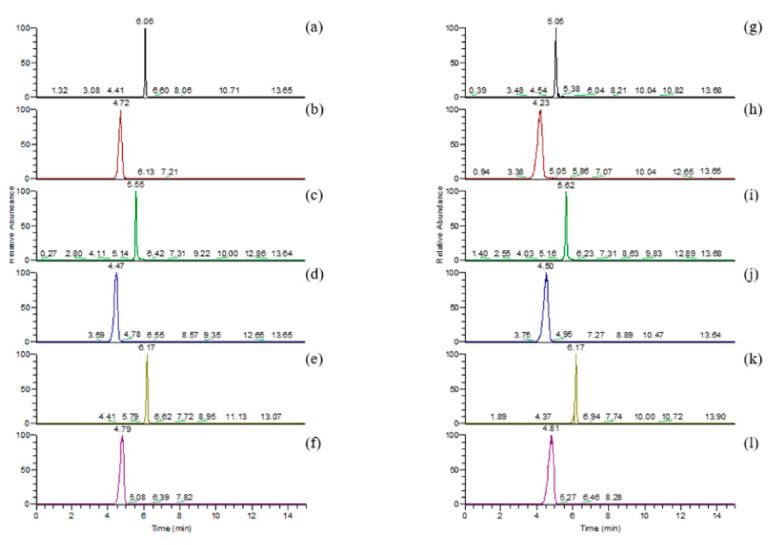
LC-MS chromatogram of anthocyanin standard. (**a**) Pelargonidin, (**b**) pelargonidin-3-glu, (**c**) Cyanidin, (**d**) Cyanidin-3-glu, (**e**) peonidin, (**f**) peonidin-3-glu. (**g**) delphinidin, (**h**) delphinidin-3- glu, (**i**) petunidin, (**j**) petunidin-3-glu, (**k**) malvidin, (**l**) malvidin-3-glu.

**Figure 4 molecules-27-03411-f004:**
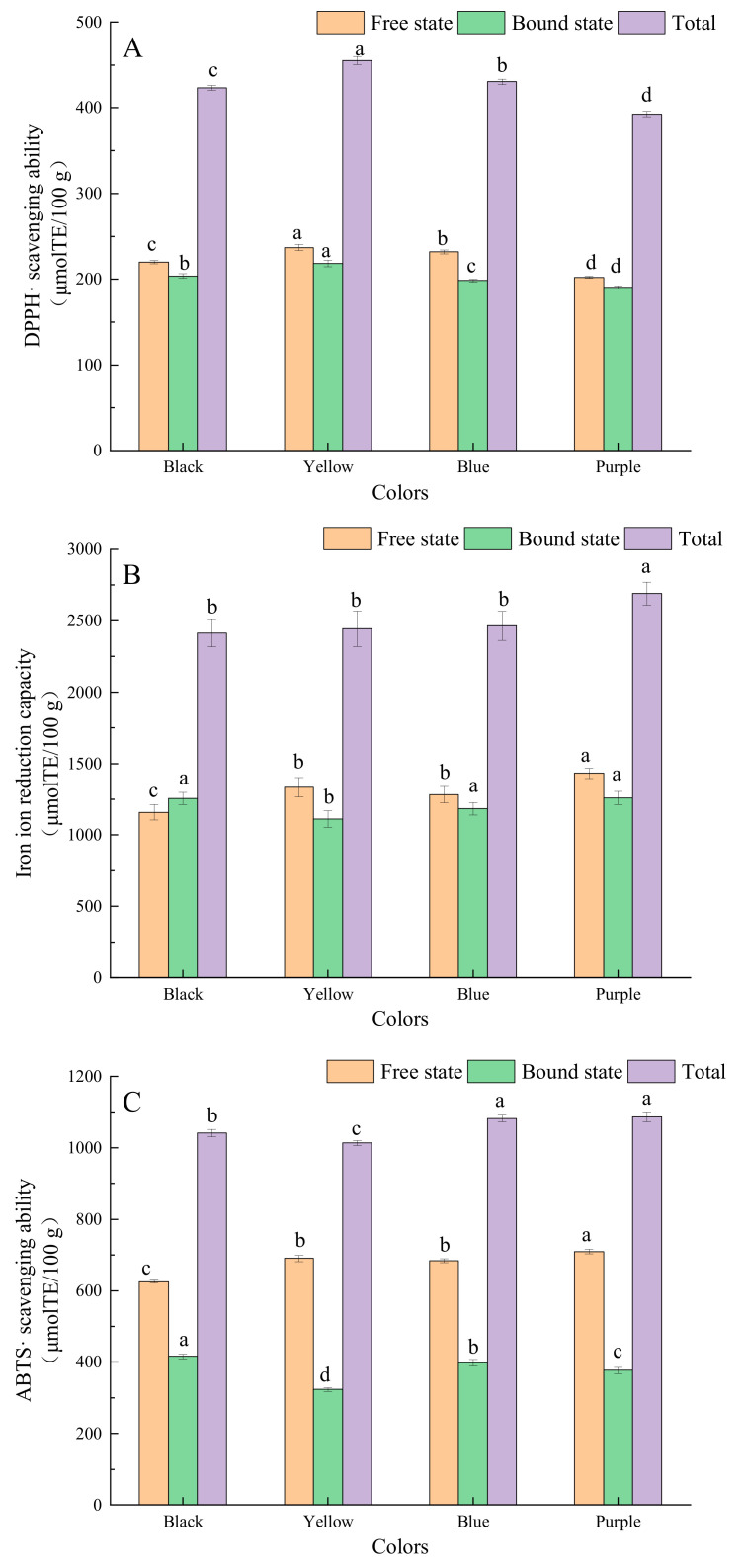
In vitro antioxidant capacity of phenolic fractions from barley of different colors. (**A**) DPPH · scavenging ability; (**B**) Iron ion reduction capacity; (**C**) ABTS · scavenging ability. Lower-case letters in the table indicate significant differences among different colors of barley about free state, bound state or total (*p* < 0.05).

**Figure 5 molecules-27-03411-f005:**
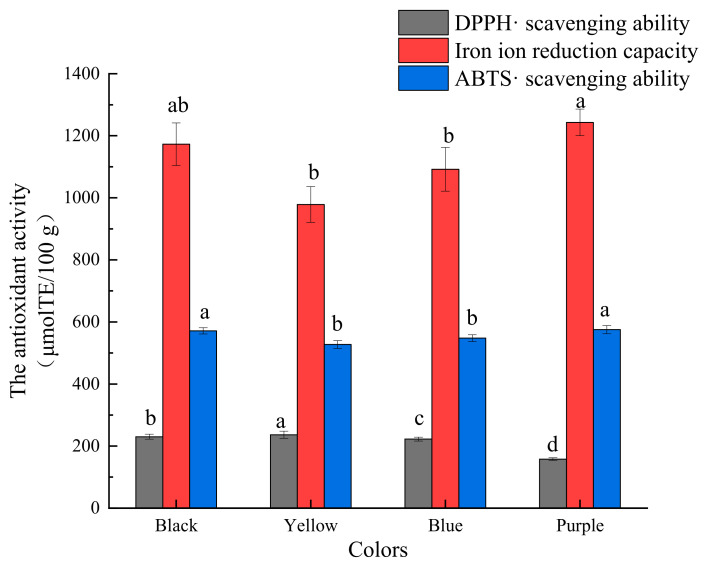
In vitro antioxidant capacity of anthocyanin fractions from barley of different colors. Lower-case letters in the table indicate significant differences among different colors of barley (*p* < 0.05).

**Figure 6 molecules-27-03411-f006:**
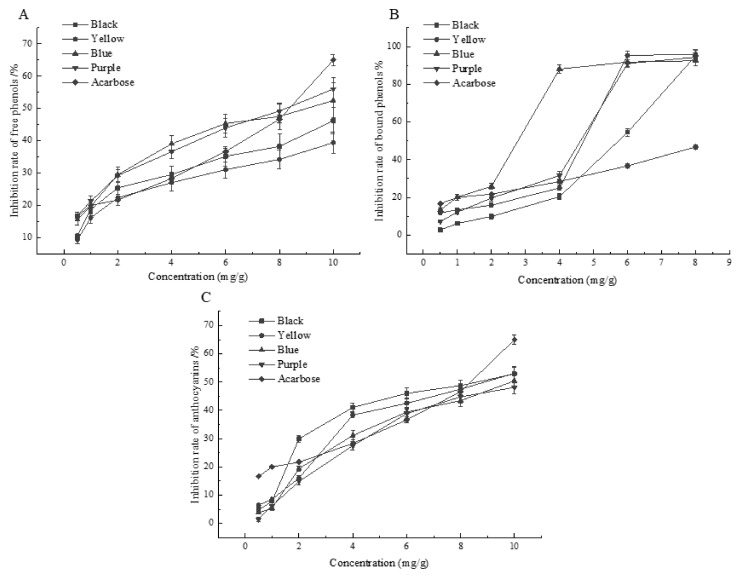
Inhibition of α-glucosidase activity in vitro by phenolic fractions of barley of different colors. (**A**) Inhibition rate of free phenolic compounds; (**B**) inhibition rate of bound phenolic compounds; (**C**) inhibition rate of anthocyanins.

**Figure 7 molecules-27-03411-f007:**
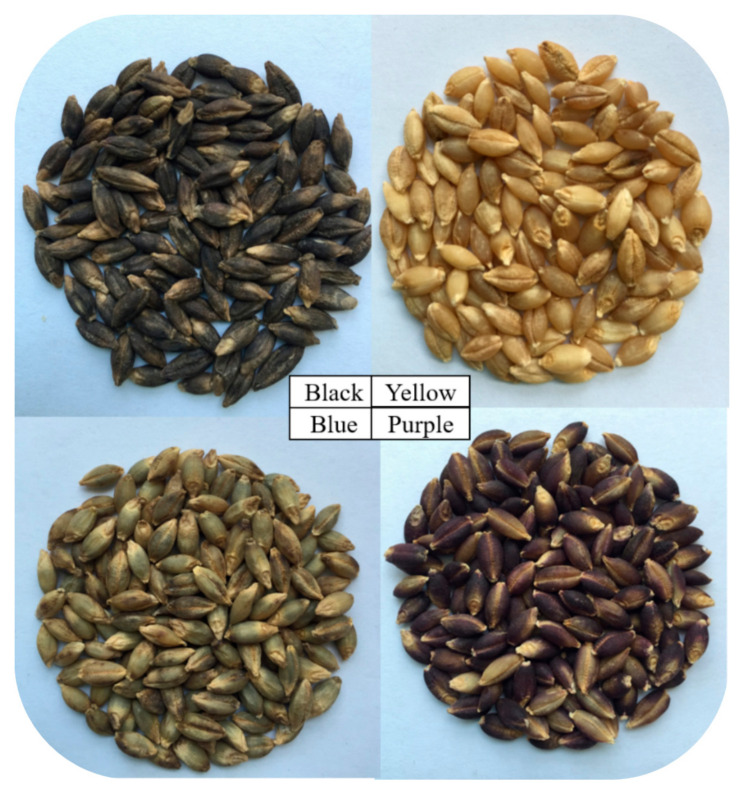
The four highland barley varieties with different colors.

**Table 1 molecules-27-03411-t001:** Content of phenolic compounds in barley of different colors (mg/100 g).

Colors	Black	Yellow	Blue	Purple
Free phenolic compounds	165.09 ± 5.17 Bc	210.32 ± 8.21 Bb	214.49 ± 8.84 Bb	264.26 ± 19.14 Ba
Bound phenolic compounds	344.35 ± 13.17 Aa	305.4 ± 13 Ab	327.79 ± 19.86 Ab	306.53 ± 5.36 Ab
Total phenolic compounds	509.44 ± 15.76 b	515.72 ± 18.57 b	542.28 ± 26.51 ab	570.78 ± 18.83 a
Free flavonoids	29.93 ± 2.64 Aa	25.87 ± 1.9 Ab	27.18 ± 2.31 Ab	27.74 ± 1.71 Ab
Bound flavonoids	15.6 ± 2 Bab	12.17 ± 2.53 Bb	11.29 ± 0.76 Bb	19.33 ± 5.64 Ba
Total flavonoids	45.53 ± 4.56 a	38.04 ± 4.34 b	38.47 ± 2.97 a	47.08 ± 6.17 a
Anthocyanins	41.91 ± 0.13 a	38.09 ± 0.06 a	33.53 ± 5.07 b	48.07 ± 8.73 a

Lower-case letters in the table indicate significant differences among different colors of barley (*p* < 0.05), whereas upper-case letters indicate those between the free and bound phenolic fractions of the same color (*p* < 0.05).

**Table 2 molecules-27-03411-t002:** Composition and content of phenolic acids and flavonoids in barley of different colors (μg/g).

Types		Black Variety		Yellow Variety		Blue Variety		Purple Variety	
		Free	Bound	Free	Bound	Free	Bound	Free	Bound
Phenolic acid									
Benzoic acids	Gallic acid	ND	ND	ND	33.365 ± 0.487 a	ND	ND	ND	15.879 ± 0.936 b
	Protocatechuic acid	4.044 ± 0.037 a	ND	1.831 ± 0.018 c	ND	2.377 ± 0.052 b	ND	ND	ND
	2,4-Dihydroxybenzoic acid	ND	ND	ND	1.458 ± 0.082 a	1.317 ± 0.049 a	ND	0.879 ± 0.024 b	ND
	Vanillic acid	2.9 ± 0.105 a	ND	ND	ND	ND	1.065 ± 0.074 a	ND	ND
	Syringic acid	ND	74.926 ± 1.016 b	ND	116.901 ± 2.056 a	ND	0.799 ± 0.014 d	ND	25.638 ± 0.745 c
	Salicylic acid	0.733 ± 0.09 b	1.3 ± 0.049 b	0.773 ± 0.046 b	1.483 ± 0.013 ab	0.847 ± 0.026 b	0.89 ± 0.033 c	1.105 ± 0.013 a	1.738 ± 0.036 a
	Benzoic acid	1.096 ± 0.015 a	84.454 ± 0.405 b	ND	140.898 ± 1.081 a	0.818 ± 0.013 b	2.699 ± 0.026 d	ND	29.596 ± 1.157 c
	Veratric acid	1.795 ± 0.033 b	11.78 ± 0.18 a	ND	0.993 ± 0.017 d	ND	4.332 ± 0.031 c	2.051 ± 0.077 a	8.442 ± 0.088 b
Cinnamic acid	O-coumaric acid	1.603 ± 0.021 a	3.614 ± 0.016 a	ND	2.267 ± 0.099 bc	ND	1.812 ± 0.039 c	ND	1.929 ± 0.024 c
	Chlorogenic acid	7.279 ± 0.26 a	ND	4.085 ± 0.162 b	ND	3.681 ± 0.165 b	6.155 ± 0.8 a	ND	ND
	p-coumaric acid	ND	11.922 ± 0.779 c	ND	14.1 ± 0.322 b	ND	583.539 ± 2.248 a	1.675 ± 0.044 a	6.912 ± 0.14 d
	Ferulic acid	ND	1.01 ± 0.023 ab	ND	1.08 ± 0.05 ab	ND	1.274 ± 0.05 a	ND	0.762 ± 0.007 b
	Total	19.450 ± 0.335 a	189.006 ± 1.323 c	6.689 ± 0.565 c	312.544 ± 2.165 b	9.040 ± 0.191 b	602.565 ± 2.097 a	5.710 ± 0.041 c	90.900 ± 1.745 d
	Total phenolic acid	208.456 ± 2.034 c		319.233 ± 1.483 b		611.605 ± 2.656 a		96.605 ± 1.023 d	
Flavone									
Hesperidin	Naringin	ND	0.668 ± 0.015 c	ND	8.066 ± 0.115 a	0.794 ± 0.005 b	6.178 ± 0.104 b	1.832 ± 0.036 a	1.131 ± 0.029 c
Flavanol	Catechin	6.458 ± 0.145 a	3.225 ± 0.097 a	5.693 ± 0.198 b	ND	ND	ND	ND	3.53 ± 0.373 a
Chalcone	Phloretin	33.078 ± 0.164 a	1.047 ± 0.073 b	ND	ND	ND	ND	1.324 ± 0.078 b	3.895 ± 0.147 a
Flavanone	Hesperidin	ND	5.564 ± 0.26 b	ND	0.704 ± 0.005 d	ND	2.002 ± 0.065 c	2.856 ± 0.074 a	9.688 ± 0.059 a
	Naringenin	0.724 ± 0.036 a	5.31 ± 0.126 c	0.616 ± 0.05 b	4.728 ± 0.066 c	0.625 ± 0.007 b	50.145 ± 1.156 a	0.698 ± 0.018 b	8.547 ± 0.151 b
Flavonol	Myricetin	ND	0.992 ± 0.029 c	ND	3.851 ± 0.049 b	ND	8.38 ± 0.737 a	ND	0.672 ± 0.005 c
	Quercetin	ND	3.084 ± 0.045 c	1.798 ± 0.091 b	8.055 ± 0.129 b	ND	24.798 ± 1.168 a	8.356 ± 0.977 a	3.729 ± 0.109 c
	Kaempferol	1.396 ± 0.485 b	3.499 ± 0.044 b	1.407 ± 0.015 a	2.602 ± 0.028 c	ND	6.991 ± 0.179 a	1.404 ± 0.059 a	2.037 ± 0.002 c
	Rutin	1.086 ± 0.267 c	4.666 ± 0.726 b	1.655 ± 0.096 b	3.769 ± 0.045 c	1.288 ± 0.078 c	5.241 ± 0.884 a	3.287 ± 0.504 a	3.22 ± 0.303 c
	Total	42.742 ± 0.986 a	28.054 ± 0.307 d	11.169 ± 0.064 c	31.775 ± 1.016 c	2.707 ± 0.044 d	103.735 ± 1.017 a	19.756 ± 1.107 b	36.450 ± 1.152 b
	Total flavone	70.796 ± 0.898 b		42.944 ± 0.906 d		106.442 ± 1.06 a		56.206 ± 0.336 c	
	Total	62.192 ± 2.632 a	217.06 ± 1.614 c	17.858 ± 0.161 c	344.319 ± 2.13 b	11.747 ± 0.999 d	706.3 ± 2.145 a	25.466 ± 1.215 b	127.346 ± 2.388 d
	Total phenol + total flavonoids	279.252 ± 2.426 c		362.178 ± 2.208 b		718.047 ± 3.021 a		152.811 ± 1.103 d	

ND, not detected. Lower-case letters in the table indicate significant differences among different colors of barley in free or bound fraction (*p* < 0.05).

**Table 3 molecules-27-03411-t003:** Composition and content of anthocyanins in barley of different colors (μg/g).

Types	Black Variety	Yellow Variety	Blue Variety	Purple Variety
Cyanidin	0.630 ± 0.01 b	0.810 ± 0.002 b	1.647 ± 0.057 a	1.772 ± 0.002 a
Cyanidin-3-glucoside	1.301 ± 0.044 c	1.267 ± 0.013 c	10.659 ± 0.088 b	96.874 ± 0.783 a
Delphinidin	1.148 ± 0.062 b	1.267 ± 0.051 b	3.269 ± 0.062 a	3.273 ± 0.076 a
Delphinidin-3-glucoside	3.013 ± 0.179 a	0.110 ± 0.074 b	3.879 ± 0.098 a	ND
Malvidin	ND	ND	ND	ND
Malvidin-3-glucoside	0.396 ± 0.004 b	ND	0.621 ± 0.039 a	0.297 ± 0.071 b
Pelargonidin	ND	ND	ND	ND
Pelargonidin-3-glucoside	0.101 ± 0.021 bc	0.104 ± 0.004 bc	0.635 ± 0.033 b	2.015 ± 0.091 a
Peonidin	ND	0.280 ± 0.006 a	0.201 ± 0.022 b	0.236 ± 0.005 ab
Peonidin-3-glucoside	ND	ND	ND	ND
Petunidin	0.405 ± 0.064 a	0.497 ± 0.019 a	0.508 ± 0.054 a	0.271 ± 0.008 b
Petunidin-3-glucoside	1.496 ± 0.025 b	0.077 ± 0.004 d	6.004 ± 0.196 a	0.475 ± 0.008 c
Total	8.492 ± 0.83 c	4.413 ± 0.442 d	27.422 ± 0.635 b	105.213 ± 2.514 a

ND, not detected. Lower-case letters in the table indicate significant differences among different colors of barley (*p* < 0.05).

**Table 4 molecules-27-03411-t004:** The table of correlation analysis.

		Iron Ion Reduction Capacity	ABTS · Scavenging Ability	DPPH · Scavenging Ability	IC50 Value for Inhibition of Alpha-Glucosidase Activity
Free phenolic fractions	Total phenolic acid content	0.996 **			
	Total cinnamic acids	0.982 *			
	Vanillic acid	0.969 *			
	o-coumaric acid	0.969 *			
	Protocatechuic acid	−0.994 **			
	Chlorogenic acid	−0.963 *			
	Phloretin	0.960 *			
Bound phenolic fractions	Gallic acid		−0.968 *		
	Benzoic acid			0.961 *	
Pelargonidin-3-glucoside					0.961 *

*. Significantly correlated at the 0.05 level (two-sided), **. Significantly correlated at the 0.01 level (two-sided).

## Data Availability

The data presented in this study are available within the article.

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
