# Peer review of "Polyphenol and Anthocyanin Composition and Activity of Highland Barley with Different Colors"

_molecules, 2022, doi:10.3390/molecules27113411_

Round 1
Reviewer 1 Report
Authors have presented a well-structured research article entitled “Polyphenol and anthocyanin contents and their composition and activity in Highland barley of different colors”. The article is original, well structured; easy to read with main emphasis on the comparative study of polyphenol and anthocyanin contents in Highland barley of different colors and their activities. In my opinion, the manuscript can be published in this journal, after the authors have addressed the following minor issues:
- Authors are suggested to improve the introduction section by adding the significance of this research.
- Authors are suggested to improve the quality of figure 1.
- Authors must provide the histogram/images of the polyphenol composition analyzed by high-performance liquid chromatography–mass spectrometry for more clarity.
- Authors are requested to add the statistical analysis in each of the figure legend.
- At few places, spelling correction and grammatical proofreading is required.
- Revised the heading of section 2 result as result and discussion.
- Authors are suggested to improve the conclusion by adding future prospective as per the standard format.
- The authors are suggested to dually check the citations throughout the manuscript.
- If possible authors are requested to incorporate a summary diagram of the complete study.
Reviewer 2 Report
Dear authors,
Your manuscript has the potential to be interesting for researchers who analyse cereal samples in generally, because it has a good methodological approach in the analysis of phenolic compounds of cereals. Moreover, this manuscript promotes applications of different coloured barley varieties, which can be potentially used as a natural source of antioxidants.
However, there are some corrections/improvements and suggestions that must be done before any further Manuscript processing.
I will enumerate different requirements to improve the manuscript. All my comments are listed below and separated as general and specific comments.
General comments:
- Title is unappropriated and confused, so it must be clarified and meaningfully rewritten.
- Abstract is too long and contains sentences which are repeating. I suggest to authors that meaningfully shorten the existing version of the abstract and to point out on the specifics of this research.
- I suggest to authors that the term “phenols” replaced with the term “phenolic compounds” in whole manuscript. Please, correct this.
- I suggest to authors that the term “bound state” replace with term “bound phenolic fraction”; whereas term “free state” replace with term “free phenolic fraction” in whole manuscript. Please, correct this.
- Several sentences in “Results and discussion” section are confuse and too long which makes understanding of text difficult. In that sense, suggest to authors to check a whole Manuscript once again and to rephrase/rewrite the mentioned sentences.
- “Correlation analysis” subsection is unappropriated in this form. Sentences are confuse and discussion is difficult for understanding. I suggest to the authors that rewrite this subsection once again.
For easier interpretation of the results, I suggest to authors that evaluate correlation only between different phenolic classes quantified by HPLC(total bound phenolic acid, total free phenolic acid, total bound flavonoid, total free flavonoid and total anthocyanins)and antioxidant/α-glucosidase activity.
- I suggest to authors that for data in Table 2 and Table 3 do the appropriate statistical analysis and determine the significance of differences between the means.
Specific comments:
Line 51-57: This sentences (“Phenols are an essential … lignans, and phenolic acids”) are confused and it must be clarified and meaningfully rewritten.
Line 62-64: This sentence (“They have a strong … and anticancer effects”) is confused and should be rewritten. Please, check and rewrite again.
Line 73-76: I suggest to authors that this sentences (“Yang reported the content … black and white”) merged into one meaningful sentence.
Line 92: Replace in sentence “…phenolic acid is the main phenolic…” with term “…phenolic acids are the main phenolic…”
Line 95-97: This sentence (“Purple barley … reported by Lin et al.”)is confused and should be meaningfully rewritten.
Line 99: Replace term “state” with term “fraction”.
Line 102-106: This sentence (“Black barley had … of other colors.”) is too long and confused. I suggest to authors that rewrite the mentioned sentences.
Line 106: Replace term “phenol” with “phenolic compounds”.
Line 113: … were not statistically significant.
Line 130: Replace term “state” with term “fraction”.
Line 154-158: This sentences (“However, these findings … of monomeric polyphenols.") are confused and unappropriated in this sections. I suggest to the authors that delete mentioned sentences, because they are not relevant to the discussion.
Line 160: …for 97.52% of the total phenolic acid content.
Line 173-174: Replace term “phenol” with “phenolic compounds”.
Line 178: Replace term “state” with term “fraction”.
Line 181: Replace term “bound extract in barley…” with “bound phenolic fraction…”
Line 182: Replace term “free extract…” with “free fraction…”
Line 211-217: This sentence (“These results indicate … the purple variety.”) is too long and confused. I suggest to authors that meaningfully rewrite the mentioned sentences.
Line 227-228: “… main types of anthocyanins in barley, all present in the four barley varieties.” replace with “…main types of anthocyanins in all barley varieties.”
Line 232: Replace term “substance” with “compounds”
Line 233-234: “…whereas peonidin was only undetected in the black variety” replace with “…while peonidin was not detected in the black variety.”
Line 239: Replace term “substance” with “compounds”
Lin 241: Replace term “was only undetected…” with term “was not detected…”.
Line 241-246: This sentence “The results of this … in the black variety.” is long and confused. Please, check and rewrite again.
Line 247: Replace term “substance” with “compounds”
Line 248-249: This sentences (“Therefore, the yellow … richer in anthocyanins.”) are confused and it must be clarified and meaningfully rewritten.
Line 257: Replace “antioxidant systems” with “antioxidant assays”
Line 259: Replace term “substance” with “compounds”
Line 263-264: This sentence (“Except for the … the bound extract.”) is confused and must be meaningfully rewritten.
Line 275: Replace term “substance” with “compounds”
Line 276: “… in the same units of wheat and oats.” Term “same units” is unappropriated in this sentence, an must be replace with adequate term.
Line 282: “… indicate on differences in …”
Line 282: Replace term “substance” with “compounds”
Line 295: Replace “…antioxidant capacities in different systems…” with “… antioxidant activity, depends of applied antioxidant assays.”
Line 430: Replace term “ methanol-fixed to …” with “dissolved in 10ml methanol, to obtain…”
Line 446: Replace term “After the …” with “Afterthat…”
Line 449: Replace term “ water-fixed to …” with “dissolved in 10 ml water and filtered through …”
Line 453-454: Replace term “phenol” with “phenolic compounds”.
Line 456: Replace “foline-phenol” with “Folin-Ciocalteu reagents”
Line 458-460: This sentence (“After 1.5h storage at room … blank solution for zeroing.”) is confused and unappropriated in this form. I suggest to authors that rewrite mentioned sentences.
Line 461: This part of sentence “… and the above procedures were repeated three times” is unappropriated in this form and must be rewritten
Line 469-472: The same comments as Line 458-460 and line 461.
Line 489-492: I suggest to the authors that describe the HPLC quantification of phenolic compounds in detail. Moreover, for quantified phenolic compounds some validation dates are missing such as: linearity, recovery, LOD, LOQ etc.
Line 497: The same comments as line 458-460.
Line 504-506: This sentences “Then, the mixed solution … for zeroing.”) is confused and must be rewritten.
Line 597: Term “DPPH” replace with term “FRAP”.
Line 512-514: The same comments as Line 504-506
Line 529-530: For which concentration of free and bound fraction the IC50 values were estimated?
Line 533: Term “expressed” replace with term “present”
Line 540: Replace term “state” with term “fraction”.
Line 551: Replace term “state” with term “fraction”.
Reviewer 3 Report
Dear Authors, dear Editor,
The submitted draft article “Polyphenol and anthocyanin contents … molecules-1723701-peer-review-v1” reports a characterization of the levels and composition of polyphenols (mostly as phenolic acids) and anthocyanin compounds in coloured cultivars and wild types of the cereal barley of the Tibetan Plateau. The reported analytical methods used are spectrophotometric measurements on the extracts, and some functional antioxidant test are performed on the extracts; also a-glucosidase inhibition assay is reported. However, Tables 2 and 3 report the composition of several compounds for the classes of acids and antocyanins, respectively. It is conceivable that these measurements were obtained by HPLC or other separation techniques, that are not reported in the Materials and Methods section, even in brief (3.2.5 refers to ref. 43, that I could download for information). (A photo of the tested seeds would be appreciated by several readers).
“This study suggested that colored barley can be used as a natural source of antioxi-559 dants and inhibit postprandial blood glucose elevation.” This statement likely comes from the glycosidase inhibition in vitro test: is there any in-vivo evidence? If yes, please refer; if NO, please state the point.
This is the only point that I would suggest to improve the article’s contents.
Best regards
Round 2
Reviewer 2 Report
I have no further comments.